# Gardeners' Past Gardening Experience and Its Moderating Effect on Community Garden Participation

**Jae Ho Lee \*** and **David Matarrita-Cascante**

Department of Recreation, Park and Tourism Sciences, Texas A&M University, College Station, TX 77843, USA; dmatarrita@tamu.edu

\* Correspondence: jaeholee83@tamu.edu; Tel.: +1-979-571-9504

**Abstract:** Studies on participation in community gardens have revealed that gardeners' participation is driven by functional and emotional motives. Most studies, however, have failed to recognize gardeners' diverse characteristics. To fill this research gap, this study examined the moderating effect that variations within gardeners has on their participation, particularly as in the case of past gardening experience. The data for this study were obtained through a survey administered in three plot-based community gardens in Austin, Texas. Results revealed that increased gardening experience bolsters the effect of emotional motivations on garden participation, while no effect was shown in the relationship between functional motivations and participation. The importance of gardeners' past gardening experience on emotional motivations is discussed as it relates to sustained participation in gardening.

**Keywords:** past gardening experience; allotment gardens; plot-based community gardens; long-term gardeners; garden participation

## 1. Introduction

Academics and practitioners have recognized the multiple benefits that community gardens produce including access to nutritious food, creating opportunities for recreation and education, enhancing community health, and developing social cohesion within communities [1–4]. (A side note: Since the types of community gardens vary (e.g., communal gardens, therapy gardens), the term 'community gardens' in this study refer to a specific garden type that consists of multiple parcels of land being 'allotted' for personal use [5]. This type of gardens is referred to as 'allotment gardens' in Europe, but we use the term community gardens as they are called community gardens in North America.) Despite the well-known benefits of gardening, the long-time survival of community gardens is under threat [6,7]. Various research findings show that the main aspects that hinder the survival of community gardens include short-term land tenure because of competing interest for development [8,9], unsecured funding from municipalities for sustaining gardens [10], and lack of participation by gardeners due to lack of gardeners' interest over time [7]. For example, according to a report conducted by the American Community Garden Association, nearly 20% of total community gardens (about 323 out of 1615) in the United States disappeared between the years 2007 to 2012 [11]. While multiple causes for their disappearance are offered (e.g., loss of land, loss of funding, loss of public/private agencies), the report stressed that the key issue hindering the longevity of community gardens is the loss of participation of gardeners over time [7,11].

To better understand ways to sustain gardeners' participation (and, subsequently, the longevity of community gardens), there have been numerous studies that have examined the motives for garden

participation. Traditionally, most of these studies have examined gardeners' functional motives, such as production of food and vegetables or enhancement of leisure opportunities [12,13]. Recently, studies [14,15] have begun to examine the emotions gardeners experience during gardening (feelings of enjoyment, psychological healing) as important motivational factors for participation in gardening. Currently, the existing body of research [16–18] is characterized by a multidimensional approach that includes functional and emotional motives.

A growing body of research [18,19] has recognized that gardeners are not a homogeneous group but rather have diverse characteristics. These include socio-economic status (mostly based on age and income), past gardening experience, activeness on gardening, the distance traveled by gardeners to gardens, and needs and motives [5,20–22]. Further, academics have recognized the link between such diversity and varying motivations for participation [23,24]. For instance, low-income gardeners have been found to be mostly driven to participate in gardening by subsistence needs due to a need for access to inexpensive food. On the other hand, upper and intermediate income gardeners were likely to participate in gardening because of a desire to protect the environment [25]. While researchers recognize that gardeners are also diverse in terms of their gardening experience, few have linked such variability to participation in gardening [20,26].

Given this gap, the present study sought to examine how past gardening experience influences garden participation. More specifically, this study sought to investigate the moderating effect that gardening experience has over functional and emotional motivational factors as they predict participation in gardening.

We expect that the study findings will contribute to the work of both academics and practitioners. Academics can benefit from expanding the body of knowledge on the effect of gardeners' different characteristics on motivations by including the impact of past gardening experience. From a practical perspective, the results of this study may provide garden managers and designers with practical implications regarding how to achieve the longevity of gardens in cities. Furthermore, along with rising concerns for food security and environmental sustainability encountered in most cities, the longevity of small green patches, assured by gardeners' sustained involvement, would contribute to maximizing disaster resilience building while at the same time encouraging smart development (e.g., edible cities, nature-based solutions).

## 2. Literature Review

### 2.1. Motivations Influencing Community Garden Participation

Studies examining the motivations leading to community garden participation [27,28] have traditionally focused on a functional perspective [12,26,29]. A functional perspective assumes that people are cognitive, have rational views, and that their participation is driven by functional aspirations for certain goals or needs [30].

One of the most prominent functional motivations to participate in gardening found in different studies is the production of food and vegetables [6,31,32]. That is, gardeners participate in gardening motivated by the need to produce food. However, it is important to note that through time, the reason for such food production has shifted from subsistence to other reasons. Differently than being motivated by subsistence reasons, modern day gardeners are interested in producing healthy and organic food [27,33]. Despite this reason, the motivation of participating in gardening for food production persists as the most important motivational factor [34,35].

In addition to food production, studies that focus on a functional perspective have reported other motivations to participation in gardening. Nordh et al. [14] and Scott et al. [36] reported leisure and recreation as another functional reason gardeners engage in gardening. Such gardeners participate in gardening in order to engage in leisure opportunities, which is particularly the case of urban settings. Similarly, other functional studies link gardeners' participation with a desire for improving physical and psychological health [26,37]. Additionally, Flachs [38] and McClintock et al. [25] noted

that reasons associated with the physical environment, such as beautification of the local environment while reducing blight, is an important source of community garden participation. Other studies have found the desire for social interaction as a functional motivator for participation in gardening [39,40].

While most studies on gardening participation have been conducted from a functional perspective, an increasing number of research has begun to recognize gardeners' emotional feelings and experiences as important motivations for garden participation [41,42]. For example, gardeners' reported feelings of joy and happiness resulting from gardening have been found to be important factors leading to participation in gardening [26,43]. Other studies following an emotional perspective have found gardeners' attachment to their garden plots as an important motivation for garden participation. For instance, Nordh et al. [14] and Lee and Matarrita-Cascante [17] noted that gardeners' emotional ties to garden plots encouraged them to continue participating in gardening, specifically in order to actively nurture and maintain their individual plots. In other words, the reviewed literature [14,27] has begun to recognize and emphasize that gardeners' emotions ascribed to their garden plots increase willingness to engage in gardening activities.

In sum, studies about community garden participation have shifted away from a unidimensional approach to a multidimensional understanding, finding that gardeners' participation is not only motivated by functional goals and needs, but also driven by intrinsic, emotional feelings that gardeners discover and experience while gardening.

## 2.2. Different Characteristics of Gardeners

Recent community gardening literature has become aware of the growing diversity of gardeners, noting that urban gardeners are not a homogeneous group [23,27]. Distinctions have been examined in the context of gardeners' socio-economic status, past gardening experience, skills and knowledge, gardening activeness, the distance traveled by gardeners to gardens, and different needs and motives [5,18,21,27]. For example, studies have found that less educated gardeners or unemployed ones are more interested in gardening for food security reasons than upper and intermediate professionals with higher education, who engage in gardening activities in order to protect the environment and produce healthy food [15,21,25]. The literature has also noted that gardeners who have been gardening for a long time are interested in producing local fruits and vegetables [44,45] and tend to express higher levels of subjective happiness and personal joy in cultivating flowers and vegetables than those who have less experience [46].

Despite studies that examined varying gardeners' characteristics, the existing body of research has not examined how these variations influence garden participation. A few exceptional studies have found that the nature of the association between motivations and garden participation is moderated by the varying characteristics of gardeners [7,20]. Specifically focusing on past gardening experience, for example, Drake and Lawson [7] noted that newer (or less experienced) gardeners tended to easily lose their interest and stop participating in gardening, while long-term (or experienced) gardeners maintained their participation. (We have to note at this point that while the notions of long-term and newer gardeners were not clearly defined in the article, such notions are used in this study as well; these notions are intended as relative terms for the purpose of comparison between gardeners by past gardening experience.) Currently, however, existing research has not yet investigated the effect of past gardening experience on individual motivational factors. To fill this gap and guide our study, we used an attitude–behavior model which allows us to examine the individual contribution of each motivational factor on garden participation as well as the moderating impact of past gardening experience on these relationships.

## 3. Framework for Analysis

To better understand the moderating impact of past gardening experience on the association between motivational factors and garden participation, this study adapted the theory of reasoned action developed by Ajzen and Fishbein [47] and modified it by including the predictor of past behavior.

(The predictor of subjective norms was omitted in our framework because, to our knowledge, no studies have found the roles of such factors in garden participation.) The theory has been widely used to analyze many different behaviors in diverse contexts when predicting people's intentional behaviors [48,49]. This theory assumes that individuals behave based on their pre-existing attitudes and behavioral intentions, thus highlighting the causal links between attitudes, intentions, and behaviors [50,51].

Important to note that the theory of reasoned action has been focused on the role of individuals' intention to behave rather than predicting behavior itself [47]. According to this theory, factors that contribute to intention to act are attitudes toward the behavior (the degree to which a person has a favorable or unfavorable evaluation of the behavior) and subjective norms (perceived social pressure to perform or not to perform a particular behavior). Here, we focused our attention on gardeners' attitudes, which are the most important factor in understanding behavioral intention [48]. Ajzen and Fishbein [47] noted that attitudes underlying a specific behavior were commonly conceptualized as two-dimensional: cognitive evaluations and affective appraisals. This study refers to cognitive evaluations as functional motivations and affective appraisals as emotional motivations.

Notably, the focus of this conceptual model is the addition of past behavior, which may influence the effects of both cognitive and affective motivations on behavior. Most of the frameworks that included past behaviors treated them as moderators when differences exist across individuals regarding their level of past behavior [52,53]. With respect to varying gardeners' characteristics on the basis of past gardening experience, this study also treated past gardening experience as a moderator, examining the influence on the relationship between motivations and gardeners' intention to participate. In particular, we tested the moderating effect of past gardening experiences to see what motivations influence garden participation according to gardeners' experience in gardening.

To avoid confusion of the impact of socio-demographics, the model includes socio-demographic characteristics as control variables, enabling a focus on the actual effects of past gardening experience. Figure 1 shows the hypothetical model for the present study, and two hypotheses were developed to guide this research:

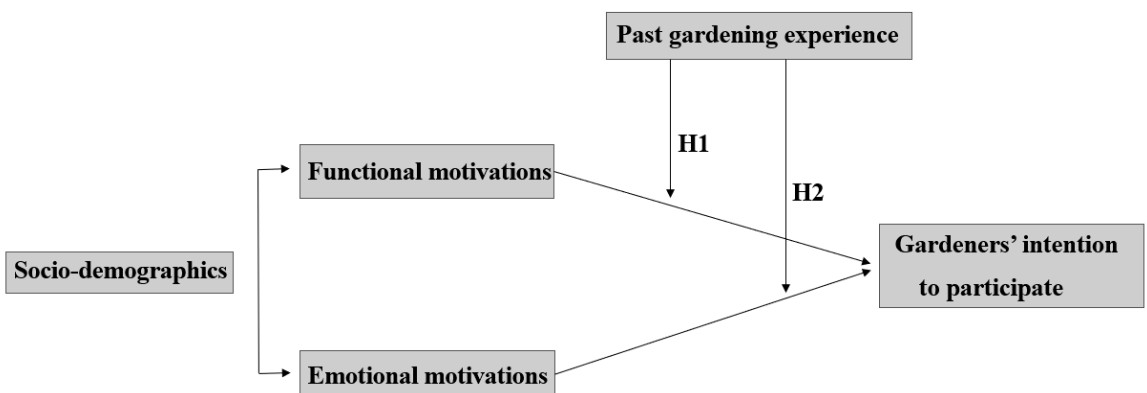

**Figure 1.** Conceptual model.

**Hypothesis 1.** *Past gardening experience moderates the effect of functional factors on gardeners' intentions to participate.*

**Hypothesis 2.** *Past gardening experience moderates the effect of emotional factors on gardeners' intentions to participate.*

## 4. Methods

### 4.1. Sampling Selection

For this study, three community gardens in Austin, Texas were selected. With a long history of urban agriculture (initiated in 1975), community gardens in Austin have served important purposes for Austin residents [54]. As they operate year-round, gardens in Austin not only produce an estimated 100,000 pounds of fresh local organic produce every year but also provide Austin's population with opportunities to meet with neighbors, restore and build their health, and explore nature within this urban area [55]. The creation and management of community gardens in Austin is mostly supported by non-profit organizations, of which the Sustainable Food Center (SFC) stands out. Specifically, non-profits in Austin provide financial support and educational programs, including technical assistance [54]. At the time of writing, 63 community gardens were in operation in Austin (see Figure 2), categorized as community gardens, communal gardens, school gardens, and therapy gardens [55].

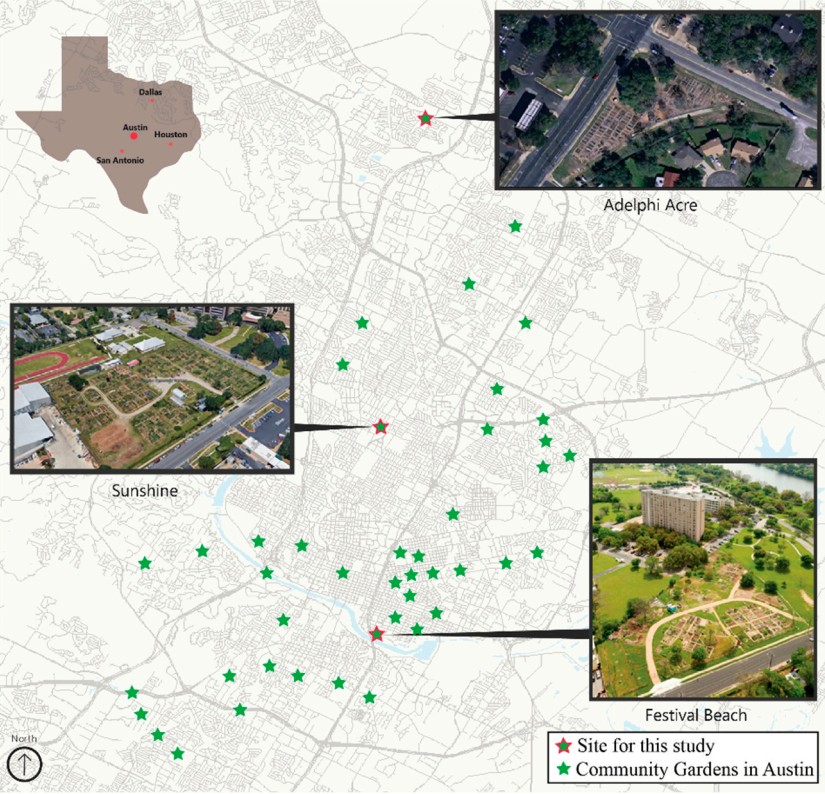

**Figure 2.** Location of the three selected gardens for this study among other public community garden sites in Austin.

The chosen gardens for this study were selected from a list of existing community gardens in Austin provided to the Principal Investigator (PI) by the Coalition of Austin Community Gardens [56]. Selection criteria for this study included: (1) gardens where participants had their own plots, qualifying them as community gardens, (2) large community gardens that had more than 50 gardeners registered, allowing for more discrete classifications between gardeners, and (3) gardens from diverse neighborhoods, improving the representativeness of gardeners' characteristics in community gardens in Austin. (A side note: To fulfill the second criterion specifically, we chose community gardens that had higher number of participants rather than prioritizing their ethnic diversity.) These selection criteria resulted in the following three gardens: Sunshine, Festival Beach, and Adelphi Acre Community Garden (see Figure 2). The details of each of them are provided in Table 1.

**Table 1.** Characteristics of the three community gardens selected for this study.

|  | Sunshine | Festival Beach | Adelphi Acre |
| --- | --- | --- | --- |
| Founded | 1979 | 2010 | 2014 |
| Number of plots | 200 | 80 | 78 |
| Registered gardeners | 270 | 90 | 94 |
| Size | 4 acres | 2 acres | 1.3 acres |
| Land ownership | School property | City of Austin | City of Austin |

*4.2. Data Collection*

The population of this study consisted of officially registered community gardeners aged 18 or older in the three chosen community gardens. The present study consisted of a three-stage process of data collection, including an elicitation study, a web-based survey (Qualtrics), and an on-site survey. These stages, conducted between 15 November 2016 and 11 March 2017, are detailed below.

In the first stage, an elicitation study was performed guided by the theory of reasoned action [47]. Since items for functional motivations vary widely based on context (e.g., food subsistence vs. environmental protection), items for functional motivations for this study were created through the elicitation study. (A side note: According to Ajzen and Fishbein [47], specific instrument items should be identified and refined from a target population because of variabilities by context, location, and the characteristics of target population.) Thus, to define appropriate items for functional motivations in the context of the three community gardens studied, 20 randomly selected gardeners (including managers, program directors, gardeners) were asked to respond to an open-ended question in November 2016: 'What do you think are the advantages (and disadvantages) for participating in community gardens?' (The most frequently mentioned functional motivations were enjoying being outdoors (20.55%) followed by accessing fresh food (17.81%), connecting with nature (16.44%), improving psychological well-being (16.44%), enhancing bodily health (13.70%), socializing with other gardeners (8.22%), and participating in social activities (6.85%)). These elicited items for functional motivations were triangulated with the literature review, and wordings and terminologies were refined by the SFC's Grow Local Program Director in Austin (Sari Albornoz). These refined items were used as a final survey questionnaire (see Supplementary Materials).

In the second stage, an online survey questionnaire was sent to gardeners in order to reach all gardeners registered in the three study gardens (Sunshine = 270, Festival = 90, Adelphi Acre = 94). Following Dillman [57], the first survey email was sent on 15 November 2016, and two reminder emails were sent in one-week intervals.

In order to capture responses from elderly gardeners who were not familiar with online surveys, the third stage involved an on-site administration of the same survey instrument utilized in the second stage of this study. This data collection stage was conducted between 14 January 2017 through 11 March 2017. The dates when the PI administered the on-site surveys were determined based on each garden's operating days (i.e., work days) when gardeners gathered the most at the garden for maintenance of communal spaces. Gardeners who agreed to participate in the study were provided with an Institutional Review Board (IRB) letter of consent describing the study in detail before beginning the survey. During this stage, interviewers left gardeners to complete the survey on their own in order to reduce interviewer bias as well as to give them more time to think [58], and instructions for drop box areas where they could leave the completed survey were provided.

In total, of the 454 respondents who were invited to participate, 191 gardeners completed a questionnaire, yielding a 42.07 percent response rate (65 percent collected via online and 35 percent collected via face-to-face surveys). The response rates of each community garden were 33.34 percent (Sunshine), 52.22 percent (Festival Beach), and 57.45 percent (Adelphi Acre). Due to inconsistent and partial responses, 11 questionnaires were deleted, resulting in 180 questionnaires being coded for analysis.

### 4.3. Measurement

Measures in the survey included functional, emotional motivations, gardeners' intention to participate in gardening, and past gardening experience. All measures included replication of indicators previously used in empirical research, except functional motivations (developed from the elicitation study).

### 4.3.1. Dependent Variable

Intention to participate in community gardens. For the variable intention to participate in community gardens, a three-item scale with a 5-point response scale (1 = strongly disagree to 5 = strongly agree) was used. The scale included statements about gardeners' level of agreement with regard to how much they were willing to keep participating in community gardening, following the guidelines by Ajzen and Fishbein [47]. Respondents were asked 'I intend to keep participating in this community garden,' 'I have decided to keep participating in this community garden,' and 'I expect to keep participating in this community garden during the next week.' An exploratory factor analysis was conducted, resulting in the extraction of single component with a Cronbach's alpha value of 0.84.

### 4.3.2. Independent Variables

Functional motivations. Measures of functional motivations for garden participation included seven Likert scale items: (1) accessing fresh food, (2) enjoying being outdoors, (3) enhancing bodily health, (4) improving psychological well-being, (5) socializing with other gardeners, (6) participating in social activities, and (7) connecting with nature. For all these items, a question was asked: 'How important is [item here] by participating in this garden to you?' Responses ranged from 1 = extremely unimportant to 5 = extremely important. An exploratory factor analysis revealed four principal components that were named as social interaction, outdoor, health, and food. Except one item for food, alpha values included 0.955 for social interaction, 0.671 for outdoor, and 0.651 for health, which were found to have acceptable internal consistency (alpha ≥ 0.6).

Emotional motivations. This study used a place attachment construct to measure gardeners' emotional motivations, as previous research has linked emotional motivations with gardeners' attachment to their plots [14,17]. Specifically, the place attachment construct developed by Kyle, Graefe, and Manning [59] was used, which consists of eight Likert scale items and has been widely adapted and used in a variety of empirical work [60,61]. The place attachment construct consists of four items measuring place identity and four measuring place dependence. In terms of the former one, items included (a) 'this garden means a lot to me,' (b) 'I am very attached to this garden,' (c) 'I strongly identify with this garden,' and (d) 'I have special connections to this garden and the people who visit it.' The four items measuring place dependence were (e) 'I enjoy visiting this garden more than any other gardens,' (f) 'I get more satisfaction out of visiting this garden than from any other garden,' (g) 'visiting this garden is more important than visiting any other gardens,' and (h) 'I would not substitute other gardens for the activities I do here.' Responses ranged from 1 (strongly disagree) to 5 (strongly agree). An exploratory factor analysis revealed two dimensions: place identity and place dependence with alpha values of 0.850 and 0.836, respectively.

Socio-demographic variables. Respondents' demographic information was collected, including gender, age, highest level of education completed, and annual household income. Table 2 shows the details of each of these socio-demographic variables.

**Table 2.** Socio-demographic profile of respondents.

| Variables | Frequency (*n*) | Percent (%) | Austin Population % [1] |
|---|---|---|---|
| Gender | | | |
| Male | 61 | 33.9 | 50.7 |
| Female | 119 | 66.1 | 49.3 |
| Age | | | |
| 19–29 | 13 | 7.2 | 19.1 |
| 30–39 | 33 | 18.3 | 19.4 |
| 40–49 | 39 | 21.7 | 13.8 |
| 50–59 | 45 | 25.0 | 10.2 |
| 60–69 | 39 | 21.7 | 7.3 |
| >70 | 10 | 5.6 | 5.5 |
| Education | | | |
| High school | 7 | 3.9 | 26.5 |
| College | 100 | 55.5 | 55.1 |
| Graduate school | 70 | 38.9 | 18.0 |
| Race | | | |
| Asian | 10 | 5.6 | 6.9 |
| Hispanic or Latino | 15 | 8.3 | 34.8 |
| White/Anglo | 132 | 73.3 | 48.5 |
| Other | 15 | 8.3 | 9.8 |
| Annual household income | | | |
| <$25,000 | 10 | 5.6 | 16.1 |
| $25,000–49,999 | 37 | 20.6 | 20.8 |
| $50,000–74,999 | 42 | 23.3 | 18.3 |
| $75,000–99,999 | 35 | 19.4 | 12.6 |
| ≥$100,000 | 37 | 20.5 | 32.1 |

[1] The percentages for the Austin population were drawn from the 2016 Census Report.

### 4.3.3. Moderating Variables

Past gardening experience. Gardeners' past gardening experience was measured with gardeners' duration of gardening. The duration of gardening was measured in the survey by asking the question: 'How long have you been gardening in this garden?' Responses ranged from less than a month to more than 6 years (see Table 3).

**Table 3.** Descriptive statistics for the duration of gardening.

| Variables | | Frequency | Percentage |
|---|---|---|---|
| | Less than a month | 8 | 4.4 |
| | 1 month to 6 months | 17 | 9.4 |
| The duration of gardening | 7 months to 1 year | 25 | 13.9 |
| | 1 to 5 years | 89 | 49.4 |
| | More than 6 years | 41 | 22.8 |

### 4.4. Data Analysis

Data analysis consisted of multiple steps in order to test the two hypotheses of this study. First, the profile of respondents was examined and compared to the demographics of the overall Austin population. This was done in order to see if participants were representative of the overall population in Austin as well as if the demographic of population was similar/dissimilar with the ones of the population in other similar studies. Second, descriptive statistics of gardeners' experience

measured by the duration of gardening were analyzed. This was conducted in order to identify a distribution of gardeners in the three community gardens studied according to their gardening experience. Third, a series of hierarchical multiple regression analyses was conducted to determine the moderating role of gardening experience on the relationships between motivations (both functional and emotional) and their intention to participate. Before conducting the regression analysis, all independent variables and moderators were mean centered (e.g., mc.x1 = x1 – mean values of x1) to prevent multicollinearity between predictor variables and interaction terms as well as to enhance interpretation of model estimates [62]. Additionally, interaction terms were generated (each of the independent variables was multiplied by past gardening experience as moderator). Then, after controlling for the socio-demographic variables, the six independent variables (mc.x1 . . . mc.x6) and the six interactions generated (mc.x1 * mc.m1 . . . mc.x6 * mc.m1) were entered into a hierarchical regression model (see Table 4). Lastly, to further examine the association between gardeners' significant interaction terms and the longevity of community gardens, regression analyses were performed by individual garden.

**Table 4.** Hierarchical regression analysis for the effects of gardening experience on the relationship between functional and emotional factors and gardeners' intention to participate.

|  | Model 1 | Model 2 | Model 3 | Model 4 | Model 5 |
|---|---|---|---|---|---|
| **Socio-demographics** | | | | | |
| Age | 0.287 *** | 0.206 ** | 0.183 ** | 0.133 * | 0.132 * |
| Gender (female) | 0.112 | 0.094 | 0.085 | 0.100 | 0.119 * |
| Education | 0.052 | −0.009 | 0.013 | −0.026 | 0.017 |
| Income | −0.052 | 0.059 | 0.066 | 0.078 | 0.045 |
| **Functional motivations** | | | | | |
| Food (mc.x1) | | 0.142 * | 0.179 ** | 0.188 ** | 0.209 ** |
| Health (mc.x2) | | 0.169 * | 0.180 ** | 0.205 ** | 0.196 ** |
| Outdoors (mc.x3) | | 0.443 *** | 0.389 *** | 0.387 *** | 0.380 *** |
| Socialization (mc.x4) | | 0.003 | 0.031 | 0.028 | 0.001 |
| **Emotional motivations** | | | | | |
| Place identity (mc.x5) | | | 0.180 ** | 0.187 ** | 0.126 * |
| Place dependence (mc.x6) | | | 0.215 ** | 0.243 *** | 0.225 *** |
| Gardening experience (mc.m1) | | | | 0.180 ** | 0.141 * |
| mc.x1 * mc.m1 | | | | | −0.058 |
| mc.x2 * mc.m1 | | | | | 0.049 |
| mc.x3 * mc.m1 | | | | | −0.004 |
| mc.x4 * mc.m1 | | | | | −0.018 |
| mc.x5 * mc.m1 | | | | | 0.159 ** |
| mc.x6 * mc.m1 | | | | | 0.143 * |
| Df | 4 | 8 | 10 | 11 | 17 |
| Adjusted R2 | 0.075 | 0.344 | 0.431 | 0.458 | 0.493 |
| F | 4.25 ** | 11.47 *** | 13.11 *** | 13.27 *** | 10.17 *** |

* $p < 0.05$ ** $p < 0.01$ *** $p < 0.001$.

## 5. Results

### 5.1. Profile of Respondents

The profile of respondents was examined and compared to the Austin population. As shown in Table 2, among a total of 180 respondents, there were 61 males (33.9%) and 119 females (66.1%), indicating that the proportion of female respondents was higher than that of male respondents compared to the latest Austin Census Report in 2016 [63]. The majority of the study respondents were aged between 50 and 59 (25.0%), including 39 respondents in both the 60–69 (21.7%) and 40–49 (21.7%) age bracket. The age distribution for the data set is older than that of the Austin population, which has

higher percentages in the age groups of 30–39 (19.4%) and 19–29 (19.1%). Regarding the level of education, nearly 40% of respondents had a graduate-level education ($n = 70$; 38.9%), which was higher than Austin's population (18%). Further, it was noteworthy that the majority of respondents were White/Anglo (73.3%), which was higher than that of the general Austin population (48.5%). Regarding household income, a significant proportion of the population (63.2%) earned more than $50,000 per year, which showed a similar distribution to Austin's population (63%).

*5.2. Descriptive Statistics for Respondents' Gardening Experience*

Table 3 presents the weighted descriptive statistics for gardening experiences as measured by duration of gardening. The duration of reported gardening showed that the majority of gardeners reported having been gardening between 1 year and 5 years (49.4%), followed by more than 6 years (22.8%), 7 months to 1 year (13.9%), 1 month to 6 months (9.4%), and less than a month (4.4%).

*5.3. Block Model Regression Analysis*

A multiple regression model was tested to investigate whether past gardening experience may moderate the effect of functional and emotional motivations on gardeners' intention to participate in community gardens.

Model 1 showed that age was statistically a significant factor influencing gardeners' intention to participate ($\beta = 0.287$, $p < 0.001$). This indicates that the older gardeners tend to participate more than younger ones. In Model 2, when functional motivations were introduced after controlling for socio-demographics, three functional variables including food ($\beta = 0.142$, $p < 0.05$), health ($\beta = 0.169$, $p < 0.05$), and outdoors ($\beta = 0.443$, $p < 0.001$), with the exception of socialization, were significantly and positively related to gardeners' intention for participation. The results show that gardeners' desires of growing food, enhancing health, and enjoying outdoor activities are associated with gardeners' intention to participate. Next, emotional motivations were introduced in Model 3 controlling for functional motivations and socio-demographics. The results showed that both place identity ($\beta = 0.180$, $p < 0.01$) and place dependence ($\beta = 0.215$, $p < 0.01$) were significantly associated with gardeners' intention to participate. This means that gardeners who have higher levels of self-identity as gardeners and who appreciated what a particular garden offers are likely to have more intention to participate in gardening. Model 4 introduced past gardening experience after controlling for socio-demographics, functional, and emotional motivations. The results of Model 4 showed that gardening experience ($\beta = 0.180$, $p < 0.01$) was significantly and positively related to gardeners' intention to participate. This means that gardeners who have more gardening experience are more likely to participate in gardening. Lastly, after controlling for the rest of the variables, all interaction terms were entered in Model 5. Results from Model 5 showed that only interaction terms generated from emotional motivations (place identity and place dependence) were significantly associated with gardeners' intention to participate ($\beta = 0.159$, $p < 0.01$; $\beta = 0.143$, $p < 0.05$, respectively). This indicates that the impact of place identity and place dependence on gardeners' intention to participate increases over time.

Lastly, to further examine the association between the significant interaction terms found in previous regression analyses (emotional motivations) and the longevity of community gardens, regression analyses by each individual garden were performed.

In the results of Table 5, both interaction terms generated from place identity ($\beta = 0.228$, $p < 0.05$) and place dependence ($\beta = 0.239$, $p < 0.05$) were statistically and positively associated with gardeners' intention to participate in the case of Sunshine Community Garden, while no significant impact was shown in the case of Festival Beach and Adelphi Acre Community Gardens. The results of this analysis mean that long-term gardeners involved in Sunshine Community Garden (the oldest) have higher levels of place identity and place dependence than gardeners in the other two gardens (relatively newly established).

**Table 5.** Hierarchical regression analysis for the effects of gardening experience on the relationship between emotional factors and gardeners' intention to participate by community garden.

|  | Sunshine (Established 1979) | Festival Beach (Established 2010) | Adelphi Acre (Established 2014) |
|---|---|---|---|
| **Socio-demographics** | | | |
| Age | 0.236 * | 0.337 * | 0.242 |
| Gender (female) | −0.076 | −0.005 | 0.433 ** |
| Education | −0.041 | 0.134 | 0.007 |
| Income | −0.038 | −0.142 | −0.053 |
| **Emotional motivations** | | | |
| Place identity (mc.x5) | 0.164 | −0.09 | 0.096 |
| Place dependence (mc.x6) | 0.309 ** | 0.389 * | 0.159 |
| Gardening experience (mc.m1) | 0.077 | 0.059 | 0.224 |
| mc.x5 * mc.m1 | 0.228 * | 0.122 | 0.275 |
| mc.x6 * mc.m1 | 0.239 * | 0.286 | −0.110 |
| Df | 9 | 9 | 9 |
| Adjusted R2 | 0.424 | 0.145 | 0.169 |
| F | 6.566 | 1.810 | 2.061 |

* $p < 0.05$ ** $p < 0.01$ *** $p < 0.001$.

## 6. Discussion

This study examined how different levels of gardening experience moderate the relationships between motivational factors (functional and emotional motivations) and gardeners' intention to participate. We hypothesized that gardening experience may moderate the effect of functional factors (H1) and emotional factors (H2) on gardeners' intention to participate in community gardens. Results showed that the hypotheses of the moderating effects on the relationships between emotional motivations and gardeners' intention to participate in gardening were fully supported, while the hypotheses of the effect of a moderator on the relationships between functional motivations on gardeners' intention for participation were not supported.

### 6.1. Insignificant Effect of Past Gardening Experience on the Association between Functional Motivations and Garden Participation

We hypothesized that past gardening experience moderates the contributions of functional factors on gardeners' intention to participate. However, the results did not support our idea, indicating that gardeners' functional motivations are not strengthened by greater past gardening experience.

The results of the study findings revealed that differences set by past gardening experience were not distinct in the effect of functional motivations determining their participation. In other words, the desires of gardeners for accessing food, seeking health, and enjoying being outdoors were common motivational factors attracting gardeners to be involved in gardening regardless of their experience in gardening. This implies that such functional motivations are gardeners' initial as well as continuing motivations leading to garden participation. Such findings are in line with previous literature [28,38], finding that gardeners' primary motivations are normally attributed to functional output (e.g., food production, personal health).

Interestingly, regardless of the levels of past gardening experience, the results of this study revealed that enjoying being outdoors was the most significant motivational factor among other functional motivations, particularly in the case of community gardens in Austin. Such findings are in line with previous research [36] that noted that most urban gardeners decided to participate in gardening due to desires for enjoying leisure activities. Furthermore, this finding affirmed findings in previous studies [23,38], which notice that gardeners who decide to be involved in gardening activities are not

exclusively motivated by a single functional motivation (traditionally defined desires for producing food and vegetables) but are driven by multiple functional motivational factors.

Lastly, it seems plausible that past gardening experience is intimately associated with a desire for socializing with other gardeners, which was not a significant functional factor in the model. This speculation stems from previous findings that gardeners develop new social ties and increase connections through interacting with neighbors in community gardens as meeting spaces [36,39]. However, the results of this study show that gardening experience does not increase gardeners' desires for socialization at the gardens. Possible explanations can be found in previous literature [1,26] that noted that community gardens (a collection of individual plots) are rather personalized places, indicating that gardeners of community gardens do not directly participate in gardening due to desires for social interaction. While social interaction between gardeners occurs within community gardens over time, motivations for social interaction do not directly contribute gardeners' intention to participate even in the case of long-term gardeners.

## 6.2. Significant Effect of Past Gardening Experience on the Association between Emotional Motivations and Garden Participation

Regarding the association between emotional motivations and gardeners' intention to participate, we found that the moderator of gardening experience facilitated the contributions of emotional factors (place identity, place dependence) on gardeners' intention to participate. This indicates that the influence of emotional factors on gardeners' intention to participate is more influential in the case of long-term gardeners than that of newer gardeners. In other words, emotional contributions to gardeners' intention to participate became stronger as gardeners are involved in gardening over time.

Such findings are noteworthy in that gardeners' emotional feelings and experiences while gardening are important factors keeping them involved in gardening. Previous studies have revealed that long-term gardeners expressed more positive emotional benefits (happiness, enjoyment) resulting from community garden participation [41,64]. Further, other studies have found that emotional aspects, such as strong attachment toward gardens, played a role in influencing garden participation [21]. Our findings further extended that such positive emotional feelings and experiences developed from garden plots over time enabled long-term gardeners to keep being involved in gardening. In other words, gardeners tend to develop their emotional attachment over time, and such developed positive emotions encourage gardeners to keep participating in gardening, leading to more regular attendance.

In this study, we investigated emotional motivations using two constructs of place attachment–place identity, place dependence. Specifically in the case of place identity, findings of previous studies have noted that self-identification with gardeners (e.g., urban farmers) played a role in influencing garden participation [14,26]. This study further revealed that such psychological connections with gardens become stronger for those who have more knowledge and skills associated with gardening. As regards place dependence, the reasons for sustained participation in the case of long-term gardeners can be attributed to their affection for their garden plots. The results can be explained by previous literature that noted that gardeners' active participation is attributed to their high levels of responsibility and obligation towards their individual plots [65,66].

Lastly, it is noteworthy that results of this study showed that long-term gardeners, especially in long-lived gardens, tend to express higher levels of emotional attachment to their gardens. This provides evidence that such strong psychological connections with gardens and affections for garden plots are associated with long-lived gardens, indicating that gardeners developed their emotional attachment through long-term engagement to a specific garden, as shown in the case of Sunshine Community Garden in this study. Most gardeners in Sunshine Community Garden, which was established in 1979 and is the longest-lived community garden in Austin, seem to have pride in and affection for their garden involvement. It seems plausible that such strong emotional attachment enables gardeners to sustain their participation in gardening; thus, we may conclude that emotional bonds or affection for garden plots may provide a path for increasing the permanency of community gardens in cities.

## 7. Conclusions and Study Limitations

This study sought to investigate how past gardening experience influences garden participation. To achieve this, the study investigated how different levels of past gardening experience moderate the associations between functional and emotional motivations and garden participation. The results of this study revealed that long-term gardening is associated with emotional attachment to gardens rather than functional motivations. Further findings showed that long-term gardeners can contribute to the longevity of community gardens, thus providing implications for allowing more stable participation in community gardens.

Practitioners can benefit from an understanding of the effect of emotional motivations as they are associated with stable garden participation as well as a contributing factor to gardens' permanency. For example, when there are threats to garden loss, commonly due to political and economic pressure in cities, long-term (experienced) gardeners, imbued by strong emotional attachment, may play an important role in helping gardens to survive, such as fighting against garden closure by organizing meetings with city council members to discuss how to sustain their gardens. In other words, community gardens cannot be sustained without long-term gardeners, and stabilizing garden participation is the key to the success of community gardens. Furthermore, such gardens with futures secured by gardeners' sense of ownership and psychological linkages toward their plots can contribute to urban resilience both directly (e.g., flood control, sustainable food production) and indirectly (e.g., urban lifestyle change and environmental education).

Despite the effective applications for garden sustainability, this study has several limitations. The first limitation in this study includes the lack of diverse characteristics of gardeners. Unlike our intentions for our site selection, a large number of gardeners who completed the survey in Austin were those who mostly earn a higher annual income and identified themselves as white/Anglo and well educated. The findings in this study may vary in other community gardener populations studied elsewhere (e.g., gardens that are made up of marginalized or financially poor participants). Secondly, the data clearly showed that more long-term gardeners completed the survey than newer gardeners in all three community gardens. Even though such a pattern is mostly observed in other community garden studies, if there was a heterogeneous sample (involving diverse groups of people), in addition to a larger sample size, the detection level depending on gardeners' different characteristics may increase to reveal more statistically significant observations.

**Supplementary Materials:** The following are available online at http://www.mdpi.com/2071-1050/11/12/3308/s1, The Influence of Emotional and Conditional Factors on Gardeners' Participation in Community Gardens.

**Author Contributions:** Data curation, investigation, methodology, J.H.L.; Supervision, D.M.-C.; Writing—original draft, J.H.L.; Writing—review and editing, D.M.-C.

**Funding:** This research received no external funding.

**Conflicts of Interest:** The authors declare no conflict of interest.

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
