# Peer review of "Gardeners’ Past Gardening Experience and Its Moderating Effect on Community Garden Participation"

_sustainability, doi:10.3390/su11123308_

Round 1

Reviewer 1 Report

The title relates to the "Gardeners past experience and its effect on community garden participation" but how did you evaluate the impact of gardener experience on emotions? If there is no experience, how can the motivation of work be increased?

the results and recommendations section should be developed.

Author Response

We would like to sincerely thank this reviewer for his/her comments and constructive suggestions that allowed us to greatly improve the quality of the manuscript. We corrected the manuscript accordingly and provided a description of each comment below.

Point 1: The title relates to the "Gardeners past experience and its effect on community garden participation" but how did you evaluate the impact of gardener experience on emotions? If there is no experience, how can the motivation of work be increased?

Response 1: We evaluated the impact of gardeners’ experience on emotions through statistical analysis, and this is a premise of our paper. In data collection, we targeted currently enrolled gardeners in the three community gardens in Austin, Texas, and gardeners’ experiences were measured ranging from less than a month to more than 6 years. This means that gardeners who did not have gardening experience were classified in the group of gardeners who have been gardening less than a month.

In regards to increasing motivations, the results of our study showed that gardeners who do not have prior gardening experience are exclusively motivated by their functional motivations. In other words, gardeners without prior gardening experience can establish their motivations by seeking access to food, health, and the outdoors, not from emotional attachment to their gardens or personal plots.

We believed that these concepts are already covered in the discussion section:

“The results of the study findings revealed that differences set by past gardening experience were not distinct in the effect of functional motivations determining their participation. In other words, the desires of gardeners for accessing food, seeking health, and enjoying being outdoors were common motivational factors attracting gardeners to be involved in gardening regardless of their experience in gardening…Regardless of the levels of past gardening experience, the results of this study revealed that enjoying being outdoors was the most significant motivational factor among other functional motivations… Findings are in line with previous research [1] that noted that most urban gardeners decided to participate in gardening due to desires for enjoying leisure activities.”

Point 2: The results and recommendations section should be developed.

Response 2: We enhanced the result section by further explaining the findings obtained in our regression analyses. Also, upon request from other reviewer, we further examined the association between the significant interaction terms (emotional motivations) and gardeners’ intention to participate by running regression analyses by individual garden. This was done to further examine the association between gardeners’ emotional motivations and the longevity of community gardens.

The results from this additional analysis showed that the impact of emotional motivations was significant in the case of Sunshine Community Garden (the oldest one), while the impact of emotional factors was not significant in the other two gardens. We added the results of regression analyses and descriptions in the result and discussion sections.

Also, based on the new findings, we revised the conclusions by stating how gardeners’ strong emotional attachment can benefit practitioners and garden managers since they are related to garden’s survival.

Reviewer 2 Report

I found the paper very well wrotten and connected with the recent directions in urban ecology. More cities proposes very ambitious actions to enhance the capacity of small green infrastructure to provide multiple benefits. For that, it is very important to understand what people expect to have and what people would like to use in urban areas. The paper needs some minor correction to improve the clarity of scientific message.

First, the paper have to be connected better with the big targets established for the cities: sustainability and resilience (e.g. SDGs) and promoting of smart development (edible cities, nature-based solutions). It is important to point out what it is the major framework for this paper.

In Introduction, the authors have to consider better the factors that influence gardeners` intentions to participate, and to introduce the political decissions, land managements, interest for developments, availability of the land, avaliability of free time, etc.

The methodological framework need few clarification. I suggest to authors to present the methodology using a flow chart. Also, in regression analysis the independent variable have to explained. Why are they relevant for analysis? One table with each variable and explanation about the relevance for explaining dependent variable could increase the clarity of the paper.     

Fig 3 and 4 are not so useful in the text.

Th discussion and conclusions (recomendations) need more connection with the results. This section need to be improved.

Author Response

We would like to sincerely thank this reviewer for his/her comments and constructive suggestions that allowed us to greatly improve our manuscript. We corrected the manuscript accordingly and provided a description of each comment below.

Point 1: First, the paper have to be connected better with the big targets established for the cities: sustainability and resilience (e.g. SDGs) and promoting of smart development (edible cities, nature-based solutions). It is important to point out what it is the major framework for this paper.

Response 1: We appreciate your compliment and suggestions to improve the quality of this paper. Upon your request, we revised the introduction and conclusion by highlighting the roles of community gardens in cities.

In the introduction, we included the text below.

“Furthermore, along with rising concerns for food security and environmental sustainability encountered in most cities, the longevity of small green patches, assured by gardeners’ sustained involvement, would contribute to maximizing disaster resilience building while at the same time encouraging smart development (e.g., edible cities, nature-based solutions).

We also included the text below in the conclusion.

“Furthermore, such gardens with futures secured by gardeners’ sense of ownership and psychological linkages toward their plots can contribute to urban resilience both directly (e.g., flood control, sustainable food production) and indirectly (e.g., urban lifestyle change and environmental education).

Point 2: In Introduction, the authors have to consider better the factors that influence gardeners` intentions to participate, and to introduce the political decisions, land managements, interest for developments, availability of the land, availability of free time, etc.

Response 2: We have added the sentence below.

“Various research findings show that the main aspects that hinder the survival of community gardens include short-term land tenure because of competing interest for development [2,3], unsecured funding from municipalities for sustaining gardens [4], and lack of participation by gardeners due to lack of gardeners’ interest over time [5].”

Point 3: The methodological framework need few clarification. I suggest to authors to present the methodology using a flow chart. Also, in regression analysis the independent variable have to explained. Why are they relevant for analysis? One table with each variable and explanation about the relevance for explaining dependent variable could increase the clarity of the paper.     

Response 3: We kindly disagree with the idea of providing a flow chart that explains the data analysis steps. However, we revisited the text in the framework section and made sure it was clear and sufficient.

In regards to adding a table explaining the relevance of each independent variable, we used the same independent variables that have been tested in a previous study (see paper by Lee & Matarrita-Cascante, 2019). According to this article, the independent variables have been identified through previous gardening literature, and the items were refined through Community Garden Program Director in Austin (we used the same sites with this study). The novelty of our paper is that we dealt with “past gardening experience” as a moderating factor, which we proved to influence gardeners’ functional and emotional motivations on garden participation. Before the current study, the roles of gardeners’ past gardening experience had not been tested yet in gardening studies.

Lee, J. H.; Matarrita-Cascante, D. The influence of emotional and conditional motivations on gardeners’ participation in community (allotment) gardens. Urban For. Urban Green. 2019, 42, 21–30.

Point 4: Fig 3 and 4 are not so useful in the text.

Response 4: We agree with this comment. After removing both graphs, we included in the text additional results of the regression analysis that show different levels of emotional attachment on garden participation between the three community gardens.

Point 5: The discussion and conclusions (recomendations) need more connection with the results. This section need to be improved.

Response 5: Upon request from other reviewer, we further examined the association between the significant interaction terms (emotional motivations) and gardeners’ intention to participate by running regression analyses by individual garden. This was done to further examine the association between gardeners’ emotional motivations and the longevity of community gardens. Thus, we have revised the discussion section based on the new findings that were shown in the hierarchical regression analysis conducted between the three gardens (Table 5 in the manuscript) analysis. Also, we revised the conclusion section to reflect the changes in the results.

Reviewer 3 Report

This is a well-written and easy to follow paper with everything laid out clearly and logically. I have three main questions/issues. Firstly, when selecting the sites and aiming for diversity, how come you did not seek or check up whether there was ethnic/income diversity between the sites - it seems an elementary oversight. Secondly, there was no comparison between the three sites, which have different durations of life as community gardens - were there any differences in place attachment, for example, was it stronger in Sunshine Garden which has been going for a lot longer than the other to? Thirdly, the the ideas for practitioners such as erecting shelters appear out of nowhere and I cannot see how the results from the data lead to these suggestions. This seems rather weak to me and to some extent devalues the work. You cannot now fix the first question but a bit more thought about the implications might be beneficial. One thing that occurs to me is that when there are threats to gardens - it was noted how there are  losses in  terms of the number of gardens - then the experienced gardeners with strong place attachment (in longer-established gardens) may be more motivated to fight against closure. Just a thought.

Author Response

We would like to sincerely thank this reviewer for his/her comments and constructive suggestions that allowed us to greatly improve the quality of the manuscript. We corrected the manuscript accordingly and provided a description of each comment below.

Point 1: Firstly, when selecting the sites and aiming for diversity, how come you did not seek or check up whether there was ethnic/income diversity between the sites - it seems an elementary oversight.

Response 1: The focus of this paper is to compare gardeners who have more gardening experience versus who have little. Because of this purpose, significant results require a large pool of participants. Therefore, we chose community gardens that had higher number of participants rather than prioritizing their ethnic diversity. Although one of our reason for considering Austin was because of the city’s ethnic diversity, respondents who participated in the survey were white/Anglo who had higher income.

Also, we included text below in a footnote in the site selection section.

“To fulfill the second criteria, we chose community gardens that had higher number of participants rather than prioritizing their ethnic diversity.”

Point 2: Secondly, there was no comparison between the three sites, which have different durations of life as community gardens - were there any differences in place attachment, for example, was it stronger in Sunshine Garden which has been going for a lot longer than the other to?

Response 2: We appreciate this comment. Upon this request, we ran a regression analysis without merging the data collected from the three community gardens regarding the significant factor (emotional motivations) found in the previous hierarchical regression analysis (Table 4). The results showed that the impact of emotional motivations was significant in the case of Sunshine Community Garden (the oldest one), while the impact of emotional factors was not significant in the other two gardens. Based on the result, we included the results of the analysis in the manuscript.

Point 3: Thirdly, the ideas for practitioners such as erecting shelters appear out of nowhere and I cannot see how the results from the data lead to these suggestions. This seems rather weak to me and to some extent devalues the work. You cannot now fix the first question but a bit more thought about the implications might be beneficial. One thing that occurs to me is that when there are threats to gardens - it was noted how there are losses in terms of the number of gardens - then the experienced gardeners with strong place attachment (in longer-established gardens) may be more motivated to fight against closure. Just a thought.

Response 3: Thank you for your suggestions to develop the quality of this paper. Upon your request, we omitted the paragraph you pointed out in the conclusion and replaced it with a new paragraph that highlighted the new findings you pointed out in Point 2.

Reviewer 4 Report

Good quality paper, presenting typical social method based study about mechanism of gardeners’ attitude to community garden participation.

Overall remarks:

Abstract is quite clear and properly describe the idea, place and topic. Methods and results are mentioned in the abstract as well, properly describing the matter and goals of the paper.

Introduction is well presented and shows the merit of a paper, only the number of disappeared gardens can be explain clearer (see “Detailed remarks” please). The second and more important think is definition of community gardening. We can find description of chosen community gardens in 4.1 (sampling selections) subsection, but better will be to include it in the Introduction section. I mean for example allotment gardens or urban gardens in courtyards of multifamily gardens. In cases like that proximity of place of living may cause deeper emotional ties, what can influence in results of survey, then it is important to define community garden or describe better chosen garden complexes. More information about different types of urban gardens, their functions, emotional feelings, experiences and motivations of gardeners Authors can find in the following book:

Urban allotment gardens in Europe / ed. by Simon Bell, Runrid Fox-Kämper, Nazila Keshavarz, Mary Benson, Silvio Caputo, Susan Noori, Annette Voigt. - London: Routledge

Literature review seems to be comprehensive study of source literature, excluding the book mentioned above, which could be enclosed to the dilatation. Especially characteristics of gardeners could be extend of explaining how far from the garden they lives.

Framework of the analysis, as well as conceptual model of research are properly designed and well shown.

Regarding methods data selection convinced me, but sometimes I felt shortage of explanation (see detailed remarks please).

Cross-section of respondents is also well verified.

Results are very interesting and well done, starting with socio-demographic profile to hierarchical regression graphs.

Discussion is comprehensive, Author referred their results very broadly, dividing discussion into two parts, and comparing them to interesting sort of cited literature.

Section number 7 – Practical Implication and Study Limitations seem to be Conclusions, but limited only to above topics. If completing it with comments about hypothesis and main results Authors can achieve good section of Conclusions.

Detailed remarks:

Introduction, first paragraph: 20% of community gardens, then how many of how many? 20% of 1615 or 1615 is 20% ?

What were the results of first open-ended question analysis?

However survey is well described regarding goals and questions, maybe it will be better to show a questionnaire? 

Author Response

We would like to sincerely thank this reviewer for his/her comments and constructive suggestions that allowed us to greatly improve the quality of the manuscript. We corrected the manuscript accordingly and provided a description of each comment below.

Point 1: Introduction, first paragraph: 20% of community gardens, then how many of how many? 20% of 1615 or 1615 is 20% ?

Response 1: We revised text as follows:

“according to a report conducted by the American Community Garden Association, nearly 20% of community gardens (about 323 out of 1,615) in the United States disappeared between the years 2007 to 2012”

Point 2: The second and more important think is definition of community gardening. We can find description of chosen community gardens in 4.1 (sampling selections) subsection, but better will be to include it in the Introduction section. I mean for example allotment gardens or urban gardens in courtyards of multifamily gardens. In cases like that proximity of place of living may cause deeper emotional ties, what can influence in results of survey, then it is important to define community garden or describe better chosen garden complexes. More information about different types of urban gardens, their functions, emotional feelings, experiences and motivations of gardeners Authors can find in the following book:

Urban allotment gardens in Europe / ed. by Simon Bell, Runrid Fox-Kämper, Nazila Keshavarz, Mary Benson, Silvio Caputo, Susan Noori, Annette Voigt. - London: Routledge

Literature review seems to be comprehensive study of source literature, excluding the book mentioned above, which could be enclosed to the dilatation. Especially characteristics of gardeners could be extend of explaining how far from the garden they lives.

Response 2: To clarify the term ‘allotment gardens,’ we edited the text in a footnote as follows:

“Since the types of community gardens vary (e.g., communal gardens, therapy gardens), the term ‘community gardens’ in this study refer to a specific garden type that consists of multiple parcels of land being ‘allotted’ for personal use [7]. This type of gardens is referred to as ‘allotment gardens’ in Europe, but we use the term community gardens as they are called community gardens in North America.”

Additionally, we added “the distance of gardens gardeners travel” when describing the diversity of gardeners in the introduction and literature review including citation as follows:

“These include socio-economic status (mostly based on age and income), past gardening experience, activeness on gardening, the distance traveled by gardeners to gardens, and needs and motives [20–23].

“Distinctions have been examined in the context of gardeners’ socio-economic status, past gardening experience, skills and knowledge, gardening activeness, the distance traveled by gardeners to gardens, and different needs and motives [18,21,23,28].”

Point 3: What were the results of first open-ended question analysis? 

Response 3: The results of the open-ended question analysis (elicitation study) below were added to the data collection section in a footnote. This is because the items gathered from the elicitation study was used as a foundation for the design of the questionnaire.

“The most frequently mentioned functional motivations were enjoying being outdoors (20.55 %) followed by accessing fresh food (17.81 %), connecting with nature (16.44 %), improving psychological well-being (16.44 %), enhancing bodily health (13.70 %), socializing with other gardeners (8.22 %), and participating in social activities (6.85 %).”

Point 4: However survey is well described regarding goals and questions, maybe it will be better to show a questionnaire? 

Response 4: We have attached a survey questionnaire that was used in data collection in the appendix.

Point 5: Section number 7 – Practical Implication and Study Limitations seem to be Conclusions, but limited only to above topics. If completing it with comments about hypothesis and main results Authors can achieve good section of Conclusions.

Response 5: In the conclusion, we included hypothesis and main results as follows:

“This study sought to investigate how past gardening experience influences garden participation. To achieve this, the study investigated how different levels of past gardening experience moderate the associations between functional and emotional motivations and garden participation. The results of this study revealed that long-term gardening is associated with emotional attachment to their gardens rather than functional motivations. Further findings showed that long-lived gardens are associated with long-term gardening, thus providing implications for allowing more stable participation in community gardens.”
